# Penetration Routes of Oxygen and Moisture into the Insulation of FR-EPDM Cables for Nuclear Power Plants

**DOI:** 10.3390/polym14235318

**Published:** 2022-12-05

**Authors:** Yoshimichi Ohki, Naoshi Hirai, Sohei Okada

**Affiliations:** 1Research Institute for Materials Science and Technology, Waseda University, 2-8-26 Nishiwaseda, Shinjuku-ku, Tokyo 169-0051, Japan; 2Oarai Research Center, Chiyoda Technol Corporation, 3681 Narita-cho, Higashi-ibaraki-gun, Oarai-machi 311-1313, Ibaraki-ken, Japan

**Keywords:** polymeric insulation, oxidation, hydrolysis, degradation, time-domain NMR

## Abstract

The polymeric insulation used in nuclear power plants (NPPs) carries the risk of molecular breakage due to oxidation and hydrolysis in the event of an accident. With this in mind, tubular specimens of flame-retardant ethylene-propylene-diene rubber (FR-EPDM) insulation were obtained by taking conductors out of a cable harvested from an NPP. Similar tubular specimens were made from a newly manufactured cable and those aged artificially using a method called the “superposition of time-dependent data.” The inner and outer surfaces of each tubular specimen were subjected to various instrumental analyses to examine their oxidation, moisture uptake, and cross-linking. As a result, it has become clear that oxygen penetrates the cable through gaps between the twisted conductor strands. Meanwhile, water vapor diffuses more often through the sheath than through gaps between the conductor strands. Of the two methods used to simulate design-based accidents in NPPs, the one used to simulate the designed loss-of-coolant accident is more severe to FR-EPDM than the one used to simulate the designed severe accident. In addition, the validity of the method called the “superposition of time-dependent data,” which is used to give artificial aging treatments to cable samples, was confirmed. Measurements of spin-spin relaxation time and residual dipolar coupling using time-domain nuclear magnetic resonance were found suitable to use to obtain information on the cross-linking of FR-EPDM insulation.

## 1. Introduction

The integrity of electric cables in charge of supplying power to devices and apparatuses and transmitting signals to control and monitor them is significant in all industries. This is, of course, also the case for nuclear power plants (NPPs). Polymer-insulated cables tend to deteriorate gradually over time, especially at NPPs, where they are exposed to heat and radioactive rays during normal operation. Some cables, called safety-related cables, are expected to perform safety functions to prevent the spread of accidents and bring them under control quickly. However, at present, the integrity of electric cables is guaranteed only by the fact that type approval was conducted upon their installation. Therefore, the long-term integrity of cables should be verified with regard to many aspects, such as through well-designed environmental performance tests.

Knowledge of the degradation mechanisms of the polymeric insulating materials used in the electric cables at NPPs should be the base for assuring the integrity of the cables. Large-scale research projects are being conducted from this viewpoint in the USA [1,2] and Europe [3,4], as well as in Japan. During the course of the Japanese project, which has been entrusted to us by the Secretariat of the Nuclear Regulation Authority in Japan, we have been conducting much research into the degradation of cross-linked polyethylene (XLPE) [5,6], flame-retardant (FR) XLPE [7], FR ethylene-propylene-diene rubber (EPDM) [8,9], FR cross-linked polyolefin [10,11], silicone rubber [12,13], and hard and soft epoxy resins [14,15,16] as typical polymeric insulating materials used in safety-related cables and electrical penetrations. As a result of such extensive research, we have found that oxidation and hydrolysis, as well as cross-linking and the breaking of chemical bonds, are fundamental degradation factors common to all of the polymers studied. This means that the penetration routes or intrusion paths of oxygen and moisture into the polymeric insulation of cables must be clarified. For the experimental verification of the integrity of polymeric cable insulation, we must also have a reliable method that can simulate the long-term changes occurring in the insulation. Here, a method called the “superposition of time-dependent data” [17,18] has been widely used in NPP-related industries. In the present research, these two areas are examined by comparing various properties between a cable harvested from an NPP and those aged artificially using the method outlined below.

## 2. Superposition of Time-Dependent Data

The superimposition of time-dependent data is a method for estimating the progress of the aging deterioration of the polymeric insulation of an electric cable in a low-dose-rate radiation environment of an NPP. This method was proposed and endorsed by the International Atomic Energy Agency (IAEA) and other organizations, such as nuclear regulatory bodies in countries such as Japan [17,18]. Assume that an insulating polymer like FR-EPDM is irradiated with ^60^Co gamma rays at a given temperature (*T*) and dose rate (*D*) for a certain aging period (*t*). When we take the elongation at break (EAB) as an aging indicator, the relation between the indicator and *t* would become an inverse stepwise-like downward shape on a semi-logarithmic graph similar to those shown in Figure 1. Here, *T*_1_, *T*_2_, and *T*_3_ are the temperatures when the EAB values were measured, while *D*_1_, *D*_2_, and *D*_3_ are the corresponding dose rates. Further, *T*_ref_ and *T*_real_ are the temperatures at which EAB was measured when *D* = 0 and in the actual environment, respectively, while *D*_real_ is the dose rate in the actual environment.

The measured values plotted on such a monotonically declining curve are called time-dependent data. Previous research has indicated that time-dependent data at a certain set of *T* and *D* can superpose nearly perfectly on those at different sets of *T* and *D* if we move them horizontally when the aging indicator is EAB. As described above, since the abscissa of Figure 1 is logarithmic, the horizontal movement of the data means that we use α*t* by multiplying *t* by the shift factor α.

Here, α is expressed as
(1)$$\alpha =\exp\left[\frac{-E}{R}\left(\frac{1}{T}-\frac{1}{T_{ref}}\right)\right]\cdot \left[1+kD^{x}\exp\left[\frac{Ex}{R}\left(\frac{1}{T}-\frac{1}{T_{ref}}\right)\right]\right]$$
where *E* and *R* are the activation energy and the gas constant, respectively, while *x* and *k* are dimensionless model parameters. Detailed explanations of Equation (1) can be found elsewhere [17,18]. Although *D* essentially has a unit such as Gy/h, it is normalized to be dimensionless in Equation (1), while *T* has the unit K. 

In Japan, the values of EAB have already been measured for most polymeric insulating materials used in safety-related cables, in both boiling and pressurized light water reactors, for each manufacturer when they were subjected to concurrent aging with heat and radiation under many combinations of *T* and *D*. Furthermore, through the analysis of such time-dependent data, *E* and the two parameters *k* and *x* have already been determined [18]. Therefore, we can calculate α for any concurrent aging with heat and radiation at *T* and *D*, or thermal aging. In other words, it is possible to estimate the EAB value that would be measured under any combination of *T* and *D*.

## 3. Samples and Experimental Procedures

### 3.1. Samples

Table 1 shows the types of cables used as samples. Cable A1 was used inside the containment vessel at Takahama Power Station Unit 1 of the Kansai Electric Power Company. Unit 1 was a pressurized water reactor and opened in November 1974. Cable A1 was in the containment vessel for 29 years from 1991 to 2020, including periods of suspension after the Fukushima-Daiichi accident occurred in March 2011, and regular inspections. The environment in the containment vessel had slightly positive air pressure and a temperature measured at 47 °C. However, during the whole period of 29 years, including the periods of inspections and suspensions, the temperature and the radiation dose rate around the cable are assumed to have been 61 °C and 0.2 mGy/h, respectively, taking into account the temperature rise due to the current flow and a safety margin [19].

Cable A1 was a cable used to drive an isolation valve, connecting it to an electrical penetration. Here, the penetration is a device that penetrates the wall of a containment vessel to connect cables inside and outside of it for conducting electricity while maintaining the confinement of radioactive materials inside [15]. Cable A1 is a three-core low-voltage cable, called an FR-PH cable in Japan, which uses FR-EPDM as an insulator and FR chlorosulfonated polyethylene as a sheath, as shown in Figure 2. Its nominal cross-sectional area is 5.5 mm^2^, and its insulation thickness is 1.0 mm. The conductor in each core consists of seven stranded wires with diameters of 1 mm.

In addition, we procured a new FR-PH cable, which is here called B0. Cable B0 was similar to A1, although the composition of additives such as the flame-retardant was unavoidably different due to the regulation of hazardous substances. The cross-section of the core and the insulation thickness were also different. B0 was a stranded wire with a nominal cross-sectional area of 2 mm^2^, consisting of seven strands with a diameter of 0.6 mm, while its insulation thickness was 0.8 mm.

Furthermore, we irradiated cable B0 with ^60^Co gamma rays at 100 °C under three different dose rates to simulate the aging degradation that cable A1 would have suffered for 29 years in an environment with a temperature of 61 °C and a radiation dose rate of 0.2 mGy/h. Based on the superimposition of time-dependent data [17,18], cable B0 was irradiated at 100 Gy/h for 1997 h, 150 Gy/h for 1431 h, and 200 Gy/h for 1126 h. These three cables are called B1, B2, and B3, as listed in Table 1.

In order to simulate the accident radiation environment during a loss-of-coolant accident (LOCA) assumed in a pressurized water reactor [20], cables A1, B1, B2, and B3 were exposed to gamma rays at a dose rate of 10 kGy/h at room temperature for 150 h to a total dose of 1.5 MGy. When the cable sample was irradiated as above, the letters R_L_ have been added at the end of the sample name, as in A1RL, for example.

After the above accident irradiation, some irradiated cable samples were exposed to saturated steam, first at 190 °C and 0.414 MPaG for 5 min, then at 150 °C and 0.414 MPaG for 175 min, and lastly at 123 °C and 0.12 MPaG until 360 h had passed, including the preceding 180 min. A chemical spray with NaOH with a pH of 8.5–10 and H_3_BO_3_ of 2000–3000 ppm was sprayed on the steam for the first 24 h. The steam vapor did not contain air or oxygen. When the cable sample was exposed to the steam vapor following the above accident irradiation, the R_L_ in the name of the irradiated sample has been replaced with L, as in A1L, for example.

Furthermore, in order to simulate the accident radiation environment during a severe accident (SA) assumed in a pressurized water reactor [21], cables A1, B1, B2, and B3 were irradiated with ^60^Co gamma rays at a dose rate of 10 kGy/h at room temperature for 50 h to a total dose of 500 kGy. When the cable sample was irradiated as above, the letters R_S_ have been added at the end of the sample name, as in A1R_S_, for example. In addition, after the above accident irradiation, some irradiated cable samples were exposed to saturated steam at 155 °C and 0.45 MPaG for 168 h. Again, the same chemical spray as above was sprayed during the first 24 h of steam vapor exposure. For cables that were exposed to this steam vapor following the accident irradiation, we have replaced the R_S_ in their names with S, as in A1S, for example.

For each of the four cables A1, B1, B2, and B3, we prepared their pristine versions and the four treated variations with the additional names R_L_, R_S_, L, and S. By adding cable B0, we had a total of 21 cables. For these cables, tubular samples were made to serve as test samples.

The purpose of this research was to examine the paths of penetration of oxygen and moisture into the insulation of FR-EPDM cables. In other words, we wanted to know the barrier effects of the cable sheath and core against the progress of insulation degradation, such as oxidation and hydrolysis. For this purpose, each core of cable, namely those listed in Table 1, as well as their derivatives exposed to gamma-ray irradiation and steam, was cut into lengths of 15 cm, and the conductor inside was extracted through a cut to the sheath with the insulation in its length direction. We thus obtained the tubular specimens. We here call the tubular specimen obtained from each cable by the same name as the cable.

If we simply showed the results of the 21 specimens on a graph, the sample dependence would not be clear. Therefore, we categorized the specimens into six groups, as listed in Table 2. The new, undegraded tubular specimen B0 is listed as No. 1, while specimen A1 of the cable harvested from the NPP and specimens B1 to B3 that received the simulated artificial aging equivalent to A1 are categorized as No. 2. Group No. 3 consists of specimens A1R_L_ and B1R_L_ to B3R_L_, which represent A1 and B1 to B3 after the simulated accident irradiation during LOCA, while No. 4 lists specimens A1L and B1L to B3L after the artificial irradiation and steam exposure to simulate the LOCA. Furthermore, No. 5 consists of A1R_S_ and B1R_S_ to B3R_S,_ which represent A1 and B1 to B3 after the simulated accident irradiation during SA, while No. 6 includes A1S and B1S to B3S after the artificial irradiation and steam exposure to simulate the SA.

### 3.2. Experimental Procedures 

We analyzed the inside and outside of the tubular specimen by instrumental analyses. First, Fourier transform mid-infrared absorption (FT-MIR) spectra were obtained through attenuated total reflection (ATR) at wave numbers from 400 to 4000 cm^−1^ using an FT/IR-6100 spectrometer (JASCO). We used a diamond ATR prism and the integration number was 80, which made the resolution 4 cm^−1^.

Next, in order to examine the adsorption and desorption of moisture on the insulation surface and the total weight change rate of the insulation, thermogravimetric differential thermal analysis (TG-DTA) was conducted on a tiny sample of 20 mg put in a Pt cell using a TG/DTA-6300 analyzer (SII), by raising the temperature by 10 °C/min from 50 to 900 °C in an Ar stream of 100 mL/min.

In addition, the degree of swelling and the gel fraction were measured using a thermal extraction method, while using toluene to obtain information on cross-linking. Specifically, after the weight, *W*1, of a stainless-steel mesh was measured in advance, the sample was put in it and the total weight, *W*2, was measured. After that, the sample, together with its mesh, was immersed in boiling toluene with a boiling point of 110.6 °C for 24 h. When the temperatures of the sample and its mesh returned to room temperature, their surfaces were wiped off, and the weight, *W*3, was measured. After that, they were vacuum-dried at 80 °C for 16 h and the weight, *W*4, was measured. Then, the gel fraction and degree of swelling were calculated as follows: Gel fraction (%) = (*W*4 − *W*1)/(*W*2 − *W*1) × 100 (2)
Degree of swelling = (*W*3 − *W*1)/(*W*4 − *W*1)(3)


Furthermore, as a new method for obtaining valuable information on cross-linking, time-domain nuclear magnetic resonance (TD-NMR) was conducted on five selected specimens using a Bruker TD-NMR Minispec mq-20 spectrometer. First, the Carr-Purcell-Meiboom-Gill (CPMG) method [22,23] was used to estimate the time constant *T*2 of the transverse relaxation (spin-spin relaxation) of the proton nuclear spins. Then, residual dipolar coupling (RDC) was also measured with the multiple-quantum (MQ) method [24,25] to confirm further the results obtained through the above CPMG method.

## 4. Results

### 4.1. FT-MIR Spectra

As a typical example of FT-MIR spectra obtained for tubular specimens, Figure 3 shows those measured on the outer surface (upper spectrum) and the inner surface (lower spectrum) of specimen B1L. As already explained, B1L was subjected to simulating aging degradation at the NPP and then exposed to the high-temperature steam in addition to the simulated irradiation during LOCA. 

The assignment of each absorption band to a specific chemical structure is listed in Table 3. In ATR measurements, the absorption intensity varies depending on the contact state of the ATR prism with the sample surface. For this reason, we decided to separate each absorption using the Lorentzian curve fitting and use the relative intensity of each absorption normalized by the integrated spectral intensity or area of aliphatic hydrocarbons, CH_al_, which is less susceptible to the specimen’s degradation, for analysis. Referring to the assignment listed in Table 3, the outer and inner surfaces of B1L contain OH groups, which suggests the penetration of moisture from the outside. It also exhibits the acid-type and ester-type C=O groups, suggesting the oxidation of FR-EPDM. 

From now on, to save space, individual measurement data will not be displayed. Instead, we will examine how each integrated spectral intensity varies between the outer and inner surfaces of each tubular specimen or among the above 21 specimens. In other words, we will look at how it changes with the progress of aging and how it changes due to exposure to radiation and steam during LOCA or SA, while distinguishing between the inner and outer surfaces of tubular specimens.

### 4.2. Carbonyl Groups

#### 4.2.1. Acid Type

The MIR absorption due to carbonyl groups is a typical indicator of the oxidation of the polymeric insulation of a cable [4,9,10,11,15]. Figure 4a shows the variation in the integrated spectral intensity due to carbonyl groups in the acid form, measured on the outer surface of each specimen among those subjected to different treatments. The outer surface of the tubular specimen is the surface of the polymeric insulation of each cable in contact with the sheath, mainly composed of FR chlorosulfonated polyethylene. 

The numbers 1 to 6 on the abscissa correspond to those listed in Table 2. For details, refer to the explanations mentioned in the paragraph above Table 2. A summary of the meanings of 1 to 6 can also be found in the figure caption. By referring to the above number and the shape and color of the data symbol listed in Table 1, we can identify each sample. For example, the green upright triangle in column four of Figure 4a represents the normalized absorption of specimen B2L. In addition, the orders 1→2→3→4 and 1→2→5→6 are the processing time sequences. However, as mentioned above, since the tubular specimen was made after each treatment had been given to the corresponding cable, the same specimen was not transferred to the next sequence. 

On the other hand, Figure 4b shows the variation of the integrated spectral intensity due to carbonyl groups in the acid form, measured on the inner surface of each specimen, among those subjected to different treatments. The inner surface of the tubular specimen corresponds to the surface of the polymeric insulation of each cable in contact with the stranded wire conductor.

Although most of the data in Figure 4a exhibit the values of normalized absorbance due to acid carbonyl around 0.01 or less, only one specimen in group No. 6 has a data point with a value of around 0.032. Likewise, specimen No. 1 exhibits a value of around 0.019 in Figure 4b. The integrated intensity of absorption due to the acid-type carbonyl groups is higher in Figure 4b on the inner surfaces of the tubular specimens in contact with the stranded wire conductors than on the outer surfaces shown in Figure 4a, in specimen A1 of the harvested cable, as well as in B1 to B3 artificially aged equivalently to A1. However, this is not the case in some samples like B1L and B1S. In this sense, the absorption due to the acid-type carbonyl groups is not adequate to examine the intrusion path of oxygen.

#### 4.2.2. Ester Type

Figure 5 shows the variation in the integrated absorption intensity due to carbonyl groups in the ester form, another indicator of the oxidation of polymeric insulation, measured on the outer and inner surfaces of each specimen subjected to different treatments. Figure 5a shows the results measured on the outer surfaces of the tubular specimens in contact with the sheaths, while Figure 5b shows the results measured on the inner surfaces in contact with the stranded wires. Slightly different from the acid-type carbonyl groups, the intensity of the ester-type carbonyl groups is higher in Figure 5b than in Figure 5a. Namely, it is higher on the inner surfaces of the tubular specimens in contact with the stranded wires than on the outer surfaces of the specimens, in all cables subjected to different treatments.

#### 4.2.3. Penetration Path of Oxygen

Figure 4a,b show that the absorption intensity due to the acid-type carbonyl groups is high in the undegraded specimen B0, but does this result indicate the oxidation of the surface of FR-EPDM insulation? To examine this, instrumental analyses were made on the surface of FR-EPDM insulation. As a result, gas chromatography-mass spectrometry (GC/MS) confirmed that the insulation contained at least two typical saturated fatty acids, namely palmitic acid and stearic acid, presumably as glidants [21]. Therefore, these two acids are likely to be misidentified as acid-type carbonyl groups present in FR-EPDM. The GC/MS analysis also indicates that the FR-EPDM insulation contains triallyl isocyanurate (TAIC) as a cross-linking agent. Here, TAIC has carbonyl groups with C and N on both sides, exhibiting an absorption band at 1690 cm^−1^ [26]. Since 1690 cm^−1^ is close to the absorption band of the acid-type carbonyl groups, it is also possible that the absorption intensity of the acid-type carbonyl groups in FR-EPDM was misidentified. 

Meanwhile, substances with ester-type carbonyl groups are not found in FR-EPDM additives. Therefore, it is appropriate to discuss the oxidation of FR-EPDM based on the intensity of ester-type carbonyl groups shown in Figure 5. As mentioned already, the intensity of the ester-type carbonyl groups is higher on the inner surfaces than on the outer ones. This result indicates that more oxygen reaches the inner surfaces, on the condition that esters and their precursors do not have a diffusion process different from that of oxygen.

Consequently, the results of Figure 4 and Figure 5 suggest that oxygen penetrates along the length direction through stranded wires in each core. That is, it is speculated that the oxygen that oxidizes the cable insulation passes into the insulation through the interstices in the strands of the core conductors, rather than through the FR chlorosulfonated polyethylene sheath. As mentioned in Section 3.1, cable A1 was installed for 29 years in a containment vessel filled with air of a slightly positive pressure. Therefore, the environmental condition around the core conductors seems to be in equilibrium with that of the containment vessel if the oxygen consumed by the oxidation of the insulator is compensated through the strands of the conductors. If this is the case, the above result, that carbonyl groups are present at the higher content on the inner surfaces, is easily understood.

### 4.3. OH Groups

As a final result of the FT-MIR spectra, Figure 6 shows the variation in integrated absorption due to hydroxyl groups among various tubular specimens subjected to different treatments. Figure 6a represents the results measured on the outer surfaces of the specimens in contact with the sheaths, while Figure 6b denotes those measured on the inner surfaces in contact with the stranded wire conductors.

In all measurements, the intensity of hydroxyl groups is much higher on the outer surface of the insulation, shown in Figure 6a, than on the inner surface, shown in Figure 6b. These results suggest that, unlike oxygen, moisture, water vapor, and humidity diffuse more easily through the FR chlorosulfonated polyethylene sheath than through the interstices in the stranded wires.

### 4.4. TG-DTA Spectra

#### 4.4.1. Desorption of Moisture

Thermogravimetric loss measured by TG-DTA in the temperature range of 50 to 150 °C corresponds to the desorption of moisture or water from the tubular specimens. Figure 7 shows the specimen dependence of the thermogravimetric loss. We can estimate from this figure the amount of water originally occluded in the FR-EPDM insulation and its variation caused by exposure to different treatments. At first glance, the sample dependence shown in Figure 7 is similar to that shown in Figure 6. Although this is a natural result, it also shows the high reliability of both measurements.

The measured values of the tubular specimen of cable A1, removed from the actual plant, and those of B1 to B3, subjected to artificial aging that seems equivalent to A1, exhibit similar dependence on the subsequent treatments. That is to say, assuming that there is a prediction that the amount of desorbed moisture will increase after steam exposure, the water desorption becomes high in A1L and B1L to B3L after the LOCA-simulated steam exposure compared with A1 and B1 to B3. However, the water desorption does not increase so much in the four specimens A1S and B1S to B3S after the exposure to steam that simulates the SA.

Regarding the steam exposure, it is difficult to judge which is more severe: the conditions simulating the LOCA, or those simulating the SA. However, the total dose irradiated was 1.5 MGy in the LOCA simulation, whereas it was 0.5 MGy in the SA simulation. We have already confirmed that FR-EPDM would not be affected and embrittled so severely, even after steam exposure that simulated a SA by conducting experiments with sheet samples, if the total irradiated dose, including the one simulating the SA, was about 0.5 to 0.9 MGy [9]. Even if that was the case, the results shown in Figure 7 indicate that the FR-EPDM insulation becomes embrittled if it is exposed to high-temperature steam after the high dose irradiation of 1.5 MGy in addition to the irradiation equivalent to the aging at the NPP. This seems to be the reason that A1L and B1L to B3L show significant moisture adsorption.

#### 4.4.2. Total Weight Loss

Figure 8 compares the total weight loss rate among the test specimens, measured over the entire heating range from 50 to 900 °C. Similar to the amount of water desorption shown in Figure 7, tubular specimen A1, made of the cable harvested from the NPP, and specimens B1 to B3, which suffered equivalent aging to A1, exhibited similar treatment dependence. In detail, A1 and B1 to B3 exhibited a lower total weight loss rate than the new specimen, B0. Although A1R_L_, which received the LOCA-simulating irradiation, showed nearly the same rate of weight loss as A1, the rate is small for B1R_L_ to B3R_L_ compared to B1 to B3. The reason for this seems to be that the progress of cross-linking caused by the aging and LOCA-simulated irradiation, and the resultant mechanical hardening, causes less weight loss. 

However, in A1L and B1L to B3L, which were exposed to the high-temperature steam in addition to the LOCA simulation, the FR-EPDM insulation became embrittled and increased the weight loss. Meanwhile, the weight loss in A1S and B1S to B3S after the SA simulation hardly increased, or even decreased, compared to A1R_S_ and B1R_S_ to B3R_S_. The embrittlement does not proceed from the steam exposure after the gamma-ray irradiation in the SA simulation. This probably reflects the fact that the total dose of gamma-ray irradiation in the SA simulation was one-third of that in the LOCA simulation, as explained in the discussion regarding Figure 7.

### 4.5. Gel Fraction and Degree of Swelling

#### 4.5.1. Gel Fraction

Figure 9 shows the changes in gel fraction among the specimens measured using the toluene immersion and heat extraction method. The gel fraction is the ratio of the mass of the polymer being tested after drying to the mass of the original polymer, as expressed by Equation (2). If the cross-linked portions of the polymer do not dissolve in the solvent and they become a gel, the gel fraction has the same meaning as the degree of cross-linking. The possibility cannot be denied that the re-organization of broken chains would result in disordered random three-dimensional structures in addition to cross-linked ones. Even so, the values on the ordinate of Figure 9 would still be a reliable cross-linking index as long as the re-organization of broken chains would result in ordered and disordered structures at a constant ratio.

The gel fractions of four tubular specimens, A1 and B1 to B3, showed very similar treatment dependence. That is to say, the gel fraction became significantly lower in A1L and B1L to B3L after the LOCA simulating treatments with irradiation and steam exposure. This suggests that the hydrolysis that occurred in these specimens induces the cutting of molecular chains and cleavage of cross-links [9].

Meanwhile, the gel fraction was higher in A1S, B1S to B3S after the SA simulation with high-temperature steam exposure. In these specimens, it is indicated that even if the molecular chains are once cut open by irradiation, they are cross-linked again by the high temperature they undergo during the steam exposure. This result adds more proof to the assumption that the high dose irradiation in the LOCA simulation causes the embrittlement of FR-EPDM, and that hydrolysis occurs in A1L and B1L to B3L when they are exposed to the high-temperature steam.

#### 4.5.2. Degree of Swelling

Figure 10 shows the treatment-dependent changes in the degree of swelling as another physical parameter measurable by the toluene immersion and heat extraction method. In general, when an organic polymer is immersed in a solvent, it absorbs the solvent and swells to expand its volume, depending on the status of the polymer. The degree of swelling is the ratio of the mass of the polymer that absorbed solvent to the mass of the polymer measured after drying in vacuum, as defined by Equation (3). Since strongly cross-linked polymers are considered less likely to swell, the degree of swelling should be inversely correlated with the degree of cross-linking or gel fraction.

Concerning the irradiation with gamma rays, we see a negative correlation between the treatment dependence of the gel fraction shown in Figure 9 and that of the degree of swelling shown in Figure 10. When the number on the horizontal axis of each graph changes in the orders of 1→2→3 and 1→2→5, the gel fraction increases, and the degree of swelling decreases. The degree of swelling decreases in 5→6 where the steam exposure of the SA simulation is involved, oppositely to the increase in gel fraction. In the process from 3 to 4 where the steam exposure of the LOCA simulation is involved, the variation of swelling degree is unclear. Except for this result, the swelling degree is inversely correlated with the gel fraction, which is consistent with the above idea.

### 4.6. NMR

Using TD-NMR, various relaxation phenomena in a substance can be studied. Among them is the transverse relaxation, observable by the CPMG method, is also called spin-spin relaxation, and its time constant is represented by *T*2 [27]. A short *T*2 corresponds to a structure with low molecular mobility or a tight-bound structure [23,27]. In this sense, *T*2 would be short in a substance with a high gel fraction, a high degree of cross-linking, and low molecular mobility. In the present research, *T*2 is divided into three components. The components with the shortest and the longest time constants are referred to as *T*2(1) and *T*2(3), respectively, while *T*2(2) represents the one in between the other two.

Figure 11 shows *T*2(1), *T*2(2), and *T*2(3) measured for five selected tubular specimens, A1, A1L, B1, B1RL, and B1L. The explanation of these five specimens has already been mentioned in many places, including in Table 1 and its footnote, but it is repeated here to increase the readability. The tubular specimens taken from the cable harvested from the NPP are A1, whereas when the irradiation and the steam exposure to simulate LOCA are given to A1, they become A1L. In addition, the tubular specimens taken from the new cable that had received the concurrent aging treatment with heat and radiation are B1. When the irradiation to partly simulate LOCA is given to B1, its samples become B1R_L_, while when both the irradiation and the steam exposure to simulate LOCA are given, they become B1L. Furthermore, since *T*2 is divided into three components with different relaxation times, *T*2(1) and *T*2(2) are respectively shown on the left and right ordinates with black circles (●) and red triangles (▲) in Figure 11(a), while *T*2(3) is shown in Figure 11(b). We put the gel fraction on the abscissae of Figure 11 since it is assumed that *T*2 and the gel fraction are negatively correlated as mentioned above. 

Based on the information above, as we see in Figure 11, *T*2(1), with the shortest relaxation time, does not exhibit a monotonic dependence on the gel fraction, but *T*2(2), with a relaxation time of about 3 ms, is clearly negatively correlated with the gel fraction. In addition, *T*2(3), with an even longer relaxation time, exhibits a negative correlation with the gel fraction. The molecular mobility in B1R_L_ should be low, since B1RL was subjected to much irradiation while simulating LOCA partly and is cross-linked. Therefore, it is reasonable that *T*2(2) and *T*2(3) are shorter in B1R_L_ than in B1. Moreover, since B1L and A1L were exposed to LOCA-simulating steam in addition to the above irradiation, molecular mobility is facilitated in them as a result of embrittlement caused by hydrolysis. Therefore, it is also reasonable that *T*2(2) and *T*2(3) are longer in A1L and B1L than in A1 and B1.

Figure 12 shows the values of RDC, *D*/2π, measured using the MQ method for five selected tubular specimens, A1, A1L, B1, B1R_L_, and B1L. It is known that the value of RDC becomes higher as topological constraints such as gelation, chemical cross-links, and physical entanglements become more significant [25,28]. Its values do not exhibit a straightforward dependence on gel fraction. However, as mentioned already, A1L and B1L should have severely deteriorated since they were treated under the simulated LOCA condition. With this in mind, if we discard A1L and B1L, RDC and gel fraction correlate positively with each other among the three specimens, A1, B1, and B1R_L_. In this sense, RDC is a good indicator of cross-links, as mentioned in [25,28].

Figure 11 and Figure 12 indicate that these TD-NMR analyses, which pay attention to relaxation and dipolar coupling phenomena, are valuable for studying the chemical structure of a polymeric insulating material and changes in its properties caused by degradation, although ordinary NMR analysis has already been widely used for such purposes [2,12,13,29,30]. 

Finally, not all the three methods for estimating the cross-linking degree of rubber, namely the toluene immersion/extraction method, the NMR-CPMG method, and the NMR-MQ method, provided consistent results. The reason for this is unclear, although we mentioned some speculations. The possibility that additives such as pigments exert some unexpected influences cannot be denied.

### 4.7. Summary of Experimental Results 

Based on the experimental results described above, we can estimate the penetration routes or intrusion paths of oxygen and water steam or moisture into the insulation of the FR-EPDM cables used at NPPs. Oxygen passes into the insulation through the interstices in the strands of the core conductors. That is, relatively speaking, the FR chlorosulfonated polyethylene sheath has a protective or barrier effect against the intrusion of oxygen. On the other hand, it is easier for water steam or moisture to diffuse through the sheath than through the interstices in strands, and the core conductors have a relatively protective effect against moisture permeation.

Furthermore, from the various experimental results shown in Figure 4, Figure 5, Figure 6, Figure 7, Figure 8, Figure 9 and Figure 10, it has become clear that various tubular specimens taken from cables, namely, A1 harvested from the NPP, B0 new and undegraded, and B1, B2, and B3 variously treated, as well as their derivatives, show reasonable and understandable treatment dependence. This indicates that the results obtained experimentally are highly reliable. Moreover, at the same time, this indicates that the superposition of time-dependent data [17,18] is highly relevant as a method of simulating the possible long-time degradation of cable insulation due to radiation and heat in a containment vessel at an NPP. However, as mentioned above, the same environmental conditions of radiation and heat were assumed during the shutdown period as those during operation in this study. We would like to reconsider this point if there is an opportunity.

## 5. Conclusions

The FR-EPDM-insulated cable installed at an NPP for 29 years was removed. A similar cable using the same insulating material was procured and degraded to the degree that seemed to be equivalent to what the removed cable would have suffered. In this simulation, a method called the “superposition of time-dependent data” was used, and a simultaneous aging treatment with heat and gamma-ray irradiation was given to the cables by changing the dose rate in three ways. Furthermore, gamma-ray irradiation and steam exposure to simulate two types of design-based accidents were added to the above cables. For tubular specimens taken from the above cables, measurements of FT-MIR spectroscopy, TG-DTA, gel fraction, swelling degree, and TD-NMR were performed on both the outer and inner surfaces. As a result, the following results and findings were confirmed.

(1) The superposition of time-dependent data for estimating the aging treatment conditions is appropriate as a tool for simulating the aging-induced changes appearing in polymeric insulation in cables installed at NPPs.

(2) As for oxygen, the FR chlorosulfonated polyethylene sheath exhibits a relative barrier effect against the intrusion of oxygen, and much oxygen enters through the interstices in the strands of the core conductors.

(3) Water steam and moisture are more likely to diffuse through the sheath than through the interstices of the wires, and the core conductor acts as a relative barrier against the moisture intrusion.

(4) The degradation of FR-EPDM after the gamma-ray irradiation and steam exposure in the LOCA simulation is more severe than that of the SA simulation, reflecting that the total irradiation dose in the former simulation is three times that in the latter.

(5) It has been demonstrated that TD-NMR analyses, which pay attention to spin-spin relaxation and dipolar coupling phenomena, are valuable for studying chemical, structural changes, such as the cross-linking of a polymeric insulating material caused by the degradation.

## Figures and Tables

**Figure 1 polymers-14-05318-f001:**
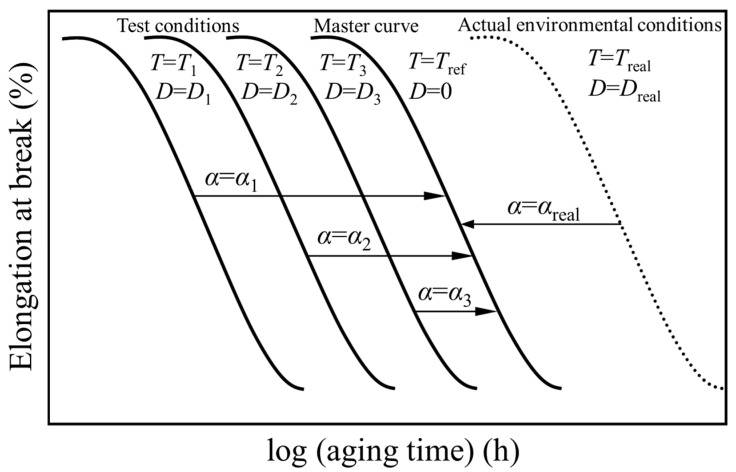
Conceptual drawing of the superposition of time-dependent data. Ordinate: elongation at break as an aging indicator. Abscissa: logarithmic aging time *t*. *T*: aging temperature. *D*: dose rate. α: shift factor.

**Figure 2 polymers-14-05318-f002:**
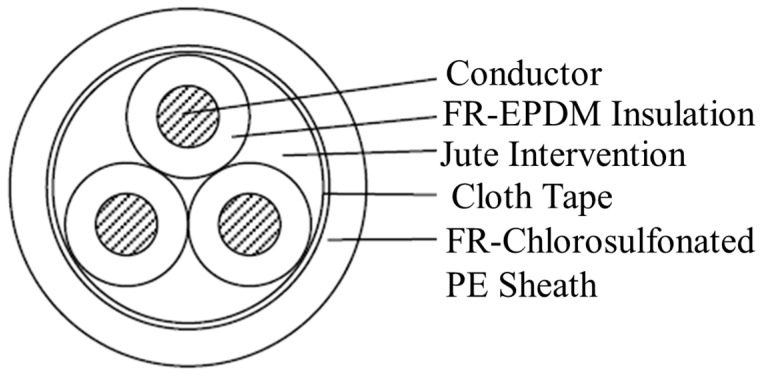
Schematic image of the cross-sections, with different sizes, of the five cables A1, B0, B1, B2 and B3 used for research. Not to scale.

**Figure 3 polymers-14-05318-f003:**
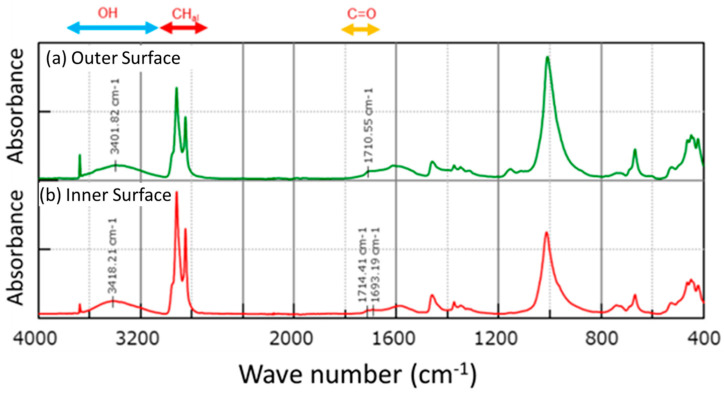
FT-MIR spectra observed on the outer surface (**a**) and the inner surface (**b**) of tubular specimens cut from cable B1L.

**Figure 4 polymers-14-05318-f004:**
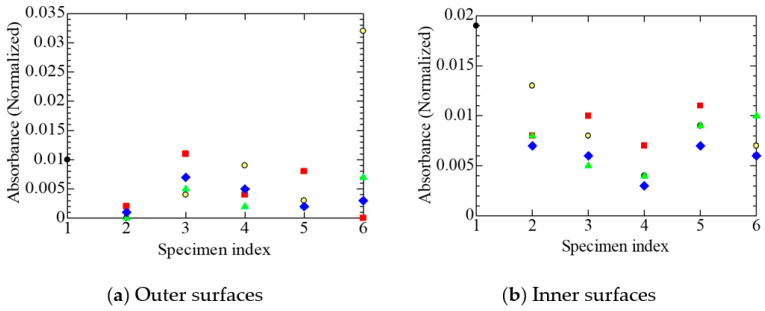
Normalized absorbance due to acid carbonyl, measured at each stage on various tubular specimens. The numeric letters on the abscissa represent the sample statuses; 1: new and unaged, 2: as removed from the NPP or aged equivalently, 3: after the radiation R_L_, 4: after the simulated LOCA, 5: after the radiation R_S_, 6: after the simulated SA. Refer also to Table 2. Refer to Table 1 for sample symbols. Several data may overlap each other.

**Figure 5 polymers-14-05318-f005:**
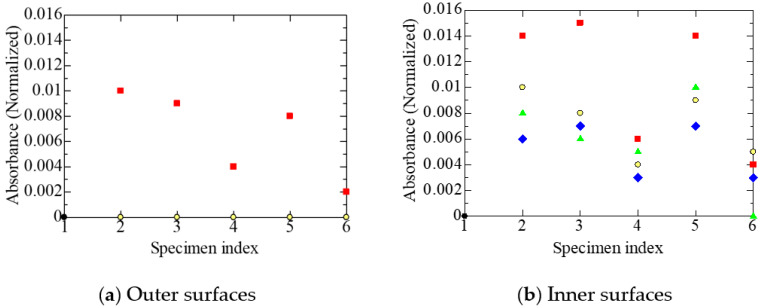
Normalized absorbance due to ester carbonyl measured at each stage on various tubular specimens. Refer to Table 1 and Table 2 and Figure 4 for the meanings of the numerals on the abscissa and sample symbols. Except for A1 and its treated stages, the absorbance in Figure 5a is null.

**Figure 6 polymers-14-05318-f006:**
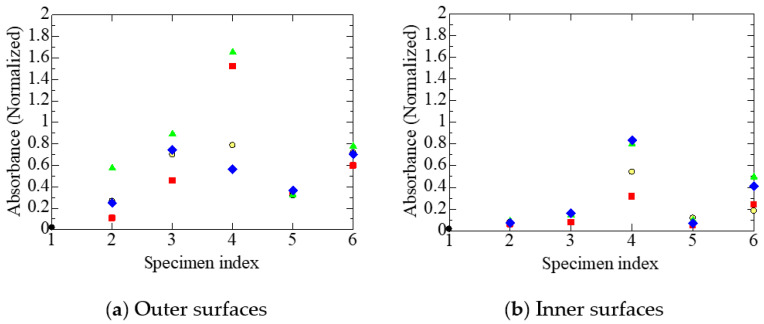
Normalized absorbance due to OH groups measured at each stage on various tubular specimens. Refer to Table 1 and Table 2 and Figure 4 for the meanings of the numerals on the abscissa and sample symbols.

**Figure 7 polymers-14-05318-f007:**
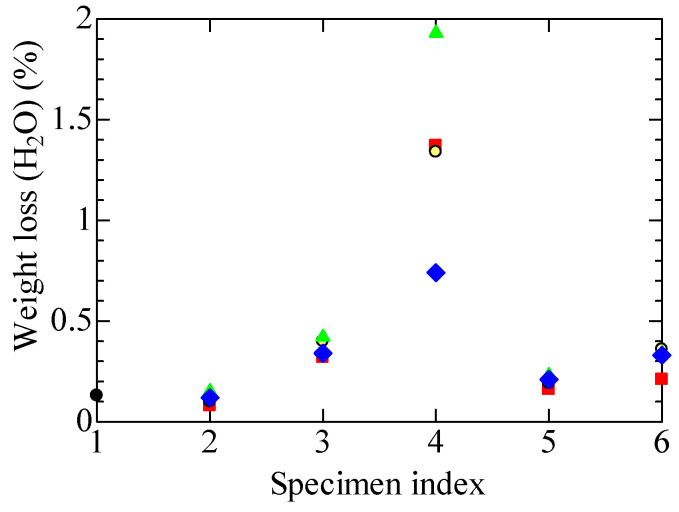
Weight losses measured at each stage in various tubular specimens due to the desorption of H_2_O in the range of 50 to 150 °C. Refer to Table 1 and Table 2 and Figure 4 for the meanings of the numerals on the abscissa and sample symbols. Several data may overlap each other.

**Figure 8 polymers-14-05318-f008:**
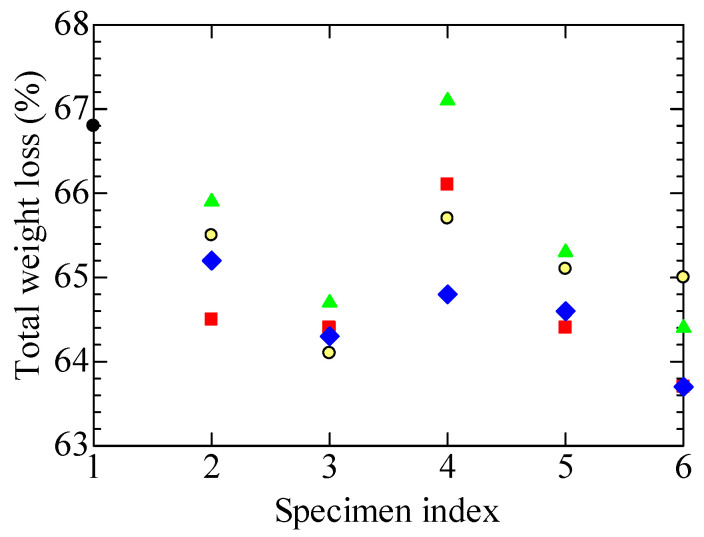
Losses in total weight in a range of 50 to 900 °C, measured at each stage for various tubular specimens. Refer to Table 1 and Table 2 and Figure 4 for the meanings of the numerals on the abscissa and sample symbols. Several data may overlap each other.

**Figure 9 polymers-14-05318-f009:**
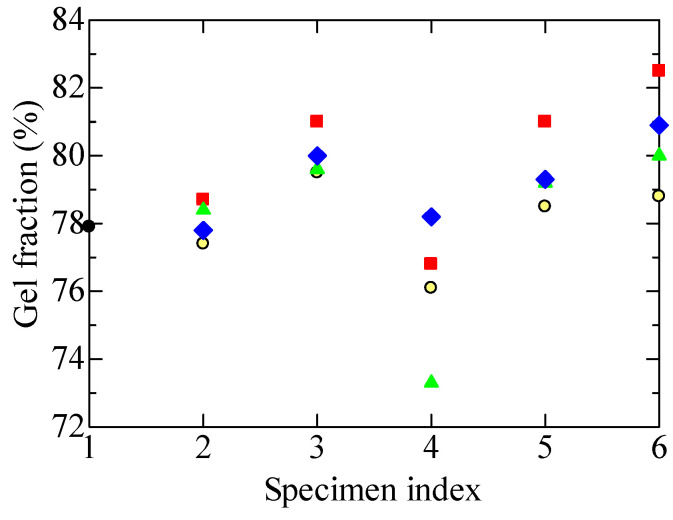
Changes in gel fraction at each stage for various tubular specimens. Refer to Table 1 and Table 2 and Figure 4 for the meanings of the numerals on the abscissa and sample symbols. Several data may overlap each other.

**Figure 10 polymers-14-05318-f010:**
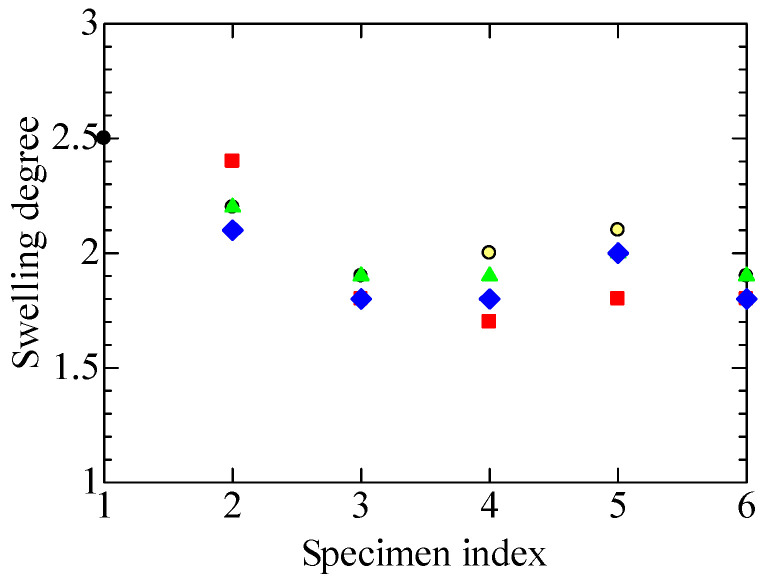
Changes in the degree of swelling at each stage for various tubular specimens. Refer to Table 1 and Table 2 and Figure 4 for the meanings of the numerals on the abscissa and sample symbols. Several data may overlap each other.

**Figure 11 polymers-14-05318-f011:**
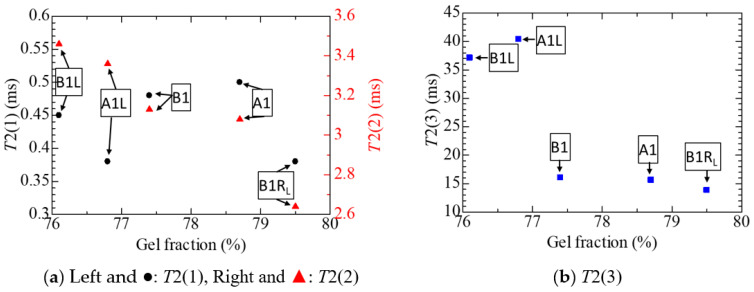
Three transverse relaxation times of the five tubular samples and their dependence on gel fraction. (**a**) ●: *T*2(1) and ▲: *T*2(2), (**b**) ■: *T*2(3).

**Figure 12 polymers-14-05318-f012:**
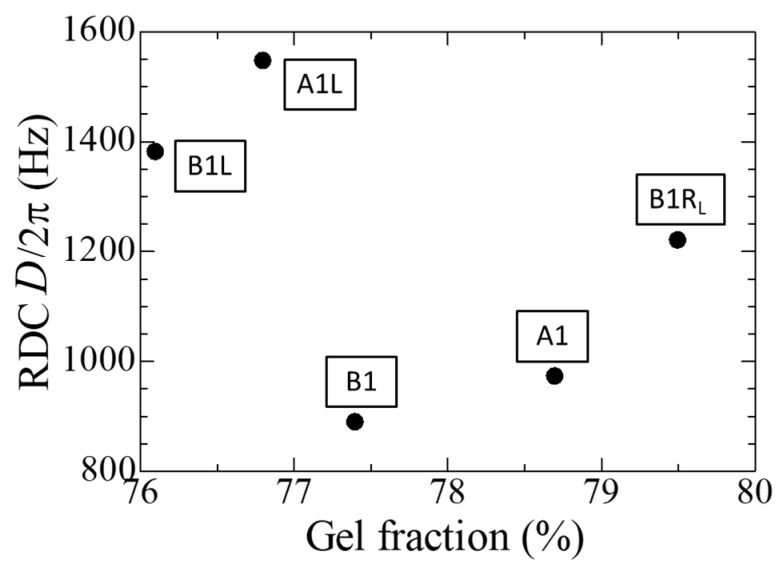
Residual dipolar coupling (RDC) measured for the five tubular samples as a function of gel fraction.

**Table 1 polymers-14-05318-t001:** Cables removed from the nuclear power plant A1, pristine unaged B0, and equivalently aged artificially B1, B2, and B3.

Sample ^1^	Gamma-Ray Irradiation
Temp.	Dose Rate	Period	Symbol
°C	Gy/h	h
A1	61	2 × 10^−4^	29 years ^2^	■
B0	-	-		●
B1	100	100	1997	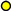
B2	100	150	1431	▲
B3	100	200	1126	◆

^1^ The letter R_L_ is added to the end of the sample names when the samples are irradiated with gamma rays at a dose of 1.5 MGy at 10 kGy/h. R_L_ is replaced with L when the samples are further exposed to high-temperature steam in the LOCA-simulating condition. The letter R_S_ is added when the gamma-ray dose is 0.5 MGy, and R_S_ is replaced with S when the samples are further exposed to steam in the severe-accident-simulating condition. ^2^ Including periods of regular inspections and suspension after January 2011.

**Table 2 polymers-14-05318-t002:** Group numbers and the specimens allocated to them.

Group Number	Specimen
1	New undegraded tubular specimen B0
2	A1 harvested from NPP and B1 to B3 artificially aged equivalently
3	A1R_L_ and B1R_L_ to B3R_L_ received LOCA-simulating irradiation
4	A1L and B1L to B3L received simulated LOCA
5	A1R_S_ and B1R_S_ to B3R_S_ received SA-simulating irradiation
6	A1S and B1S to B3S received simulated SA

**Table 3 polymers-14-05318-t003:** IR absorption bands and their assignments.

Wave Number (cm^−1^)	Assignment	Origin
3800–3200	OH	H_2_O
3000–2800	CH_al_	Aliphatic (EPDM)
around 1720	C=O_est_	Ester carbonyl
around 1700	C=O_acid_	Acid carbonyl
1460, 1380	CH_2_, CH_3_	Aliphatic (EPDM)
1010	Mg-O	Talc
720	-(CH_2_)*_n_*- *n* ≥ 4	Aliphatic (EPDM)

## Data Availability

Not applicable.

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
