# Peer review of "Penetration Routes of Oxygen and Moisture into the Insulation of FR-EPDM Cables for Nuclear Power Plants"

_polymers, 2022, doi:10.3390/polym14235318_

Round 1
Reviewer 1 Report
This manuscript studies penetration routes of oxygen and moisture
into the insulation of FR-EPDM cables for nuclear power plants.
The manuscript is publishable for the journal after some minor issues
are addressed. The detailed comments are as follows:
1. The abstract should be improved to highlight the major contribution
of the paper.
2. Regarding the format and language, the manuscript should be
checked and improved.
Reviewer 2 Report
Dear Authors,
Thank you for submitting this article to MDPI Polymers. I would say that the article is very well written and it is about a very interesting topic. Nonetheless, some minor revisions are needed and, in particular, one comment is focused on the oxygen reactions occurring during aging.
Comments and notes may be found in the attached PDF.
Regards.
